# Fast Greedy MAP Inference for Determinantal Point Process to Improve Recommendation Diversity

**Laming Chen**
Hulu LLC
Beijing, China
laming.chen@hulu.com

**Guoxin Zhang**[*]
Kwai Inc.
Beijing, China
zhangguoxin@kuaishou.com

**Hanning Zhou**
Hulu LLC
Beijing, China
ericzhouh@gmail.com

## Abstract

The determinantal point process (DPP) is an elegant probabilistic model of repulsion with applications in various machine learning tasks including summarization and search. However, the maximum a posteriori (MAP) inference for DPP which plays an important role in many applications is NP-hard, and even the popular greedy algorithm can still be too computationally expensive to be used in large-scale real-time scenarios. To overcome the computational challenge, in this paper, we propose a novel algorithm to greatly accelerate the greedy MAP inference for DPP. In addition, our algorithm also adapts to scenarios where the repulsion is only required among nearby few items in the result sequence. We apply the proposed algorithm to generate relevant and diverse recommendations. Experimental results show that our proposed algorithm is significantly faster than state-of-the-art competitors, and provides a better relevance-diversity trade-off on several public datasets, which is also confirmed in an online A/B test.

## 1   Introduction

The determinantal point process (DPP) was first introduced in [33] to give the distributions of fermion systems in thermal equilibrium. The repulsion of fermions is described precisely by DPP, making it natural for modeling diversity. Besides its early applications in quantum physics and random matrices [35], it has also been recently applied to various machine learning tasks such as multiple-person pose estimation [27], image search [28], document summarization [29], video summarization [19], product recommendation [18], and tweet timeline generation [49]. Compared with other probabilistic models such as the graphical models, one primary advantage of DPP is that it admits polynomial-time algorithms for many types of inference, including conditioning and sampling [30].

One exception is the important maximum a posteriori (MAP) inference, i.e., finding the set of items with the highest probability, which is NP-hard [25]. Consequently, approximate inference methods with low computational complexity are preferred. A near-optimal MAP inference method for DPP is proposed in [17]. However, this algorithm is a gradient-based method with high computational complexity for evaluating the gradient in each iteration, making it impractical for large-scale real-time applications. Another method is the widely used greedy algorithm [37], justified by the fact that the log-probability of set in DPP is submodular. Despite its relatively weak theoretical guarantees [13], it is widely used due to its promising empirical performance [29, 19, 49]. Known exact implementations of the greedy algorithm [17, 32] have $O(M^4)$ complexity, where $M$ is the total number of items. Han et al.'s recent work [20] reduces the complexity down to $O(M^3)$ by introducing some approximations, which sacrifices accuracy. In this paper, we propose an *exact* implementation of the greedy algorithm with $O(M^3)$ complexity, and it runs much faster than the approximate one [20] empirically.

_______________________

[*]This work was conducted while the author was with Hulu.

The essential characteristic of DPP is that it assigns higher probability to sets of items that are diverse from each other [30]. In some applications, the selected items are displayed as a sequence, and the negative interactions are restricted only among nearby few items. For example, when recommending a long sequence of items to the user, each time only a small portion of the sequence catches the user's attention. In this scenario, requiring items far away from each other to be diverse is unnecessary. Developing fast algorithm for this scenario is another motivation of this paper.

**Contributions.** In this paper, we propose a novel algorithm to greatly accelerate the greedy MAP inference for DPP. By updating the Cholesky factor incrementally, our algorithm reduces the complexity down to $O(M^3)$, and runs in $O(N^2 M)$ time to return $N$ items, making it practical to be used in large-scale real-time scenarios. To the best of our knowledge, this is the first exact implementation of the greedy MAP inference for DPP with such a low time complexity.

In addition, we also adapt our algorithm to scenarios where the diversity is only required within a sliding window. Supposing the window size is $w < N$, the complexity can be reduced to $O(wNM)$. This feature makes it particularly suitable for scenarios where we need a long sequence of items diversified within a short sliding window.

Finally, we apply our proposed algorithm to the recommendation task. Recommending diverse items gives the users exploration opportunities to discover novel and serendipitous items, and also enables the service to discover users' new interests. As shown in the experimental results on public datasets and an online A/B test, the DPP-based approach enjoys a favorable trade-off between relevance and diversity compared with the known methods.

## 2  Background and Related Work

**Notations.** Sets are represented by uppercase letters such as $Z$, and $\#Z$ denotes the number of elements in $Z$. Vectors and matrices are represented by bold lowercase letters and bold uppercase letters, respectively. $(\cdot)^\top$ denotes the transpose of the argument vector or matrix. $\langle \mathbf{x}, \mathbf{y} \rangle$ is the inner product of two vectors $\mathbf{x}$ and $\mathbf{y}$. Given subsets $X$ and $Y$, $\mathbf{L}_{X,Y}$ is the sub-matrix of $\mathbf{L}$ indexed by $X$ in rows and $Y$ in columns. For notation simplicity, we let $\mathbf{L}_{X,X} = \mathbf{L}_X$, $\mathbf{L}_{X,\{i\}} = \mathbf{L}_{X,i}$, and $\mathbf{L}_{\{i\},X} = \mathbf{L}_{i,X}$. $\det(\mathbf{L})$ is the determinant of $\mathbf{L}$, and $\det(\mathbf{L}_\emptyset) = 1$ by convention.

### 2.1  Determinantal Point Process

DPP is an elegant probabilistic model with the ability to express negative interactions [30]. Formally, a DPP $\mathcal{P}$ on a discrete set $Z = \{1, 2, \dots, M\}$ is a probability measure on $2^Z$, the set of all subsets of $Z$. When $\mathcal{P}$ gives nonzero probability to the empty set, there exists a matrix $\mathbf{L} \in \mathbb{R}^{M \times M}$ such that for every subset $Y \subseteq Z$, the probability of $Y$ is

$$\mathcal{P}(Y) \propto \det(\mathbf{L}_Y),$$

where $\mathbf{L}$ is a real, positive semidefinite (PSD) kernel matrix indexed by the elements of $Z$. Under this distribution, many types of inference tasks including marginalization, conditioning, and sampling can be performed in polynomial time, except for the MAP inference

$$Y_{\text{map}} = \arg\max_{Y \subseteq Z} \det(\mathbf{L}_Y).$$

In some applications, we need to impose a cardinality constraint on $Y$ to return a subset of fixed size with the highest probability, resulting in the MAP inference for $k$-DPP [28].

Besides the works on the MAP inference for DPP introduced in Section 1, some other works propose to draw samples and return the one with the highest probability. In [16], a fast sampling algorithm with complexity $O(N^2 M)$ is proposed when the eigendecomposition of $\mathbf{L}$ is available. Though the update rules of [16] and our work are similar, there are two major differences making our approach more efficient. First, [16] requires the eigendecomposition of $\mathbf{L}$ with time complexity $O(M^3)$. This computation overhead dominates the overall running time when we only need to return a small number of items. By contrast, our approach only requires overall $O(N^2 M)$ complexity to return $N$ items. Second, sampling algorithm of DPP usually needs to perform multiple sample trials to achieve comparable empirical performance with the greedy algorithm, which further increases the computational complexity.

## 2.2 Greedy Submodular Maximization

A set function is a real-valued function defined on $2^Z$. If the marginal gains of a set function $f$ are non-increasing, i.e., for any $i \in Z$ and any $X \subseteq Y \subseteq Z \setminus \{i\}$,

$$f(X \cup \{i\}) - f(X) \geq f(Y \cup \{i\}) - f(Y),$$

then $f$ is submodular. The log-probability function in DPP $f(Y) = \log \det(\mathbf{L}_Y)$ is submodular, as is revealed in [17]. Submodular maximization corresponds to finding a set which maximizes a submodular function. The MAP inference for DPP is a submodular maximization.

Submodular maximization is generally NP-hard. A popular approximation approach is based on the greedy algorithm [37]. Initialized as $\emptyset$, in each iteration, an item which maximizes the marginal gain

$$j = \arg \max_{i \in Z \setminus Y_\mathrm{g}} f(Y_\mathrm{g} \cup \{i\}) - f(Y_\mathrm{g}),$$

is added to $Y_\mathrm{g}$, until the maximal marginal gain becomes negative or the cardinality constraint is violated. When $f$ is monotone, i.e., $f(X) \leq f(Y)$ for any $X \subseteq Y$, the greedy algorithm admits a $(1 - 1/e)$-approximation guarantee subject to a cardinality constraint [37]. For general submodular maximization with no constraints, a modified greedy algorithm guarantees $(1/2)$-approximation [10]. Despite these theoretical guarantees, greedy algorithm is widely used for DPP due to its promising empirical performance [29, 19, 49].

## 2.3 Diversity of Recommendation

Improving the recommendation diversity has been an active field in machine learning and information retrieval. Some works addressed this problem in a generic setting to achieve better trade-off between arbitrary relevance and dissimilarity functions [11, 9, 51, 8, 21]. However, they used only pairwise dissimilarities to characterize the overall diversity property of the list, which may not capture some complex relationships among items (e.g., the characteristics of one item can be described as a simple linear combination of another two). Some other works tried to build new recommender systems to promote diversity through the learning process [3, 43, 48], but this makes the algorithms less generic and unsuitable for direct integration into existing recommender systems.

Some works proposed to define the similarity metric based on the taxonomy information [52, 2, 12, 45, 4, 44]. However, the semantic taxonomy information is not always available, and it may be unreliable to define similarity based on them. Several other works proposed to define the diversity metric based on explanation [50], clustering [7, 5, 31], feature space [40], or coverage [47, 39].

In this paper, we apply the DPP model and our proposed algorithm to optimize the trade-off between relevance and diversity. Unlike existing techniques based on pairwise dissimilarities, our method defines the diversity in the feature space of the entire subset. Notice that our approach is essentially different from existing DPP-based methods for recommendation. In [18, 34, 14, 15], they proposed to recommend complementary products to the ones in the shopping basket, and the key is to learn the kernel matrix of DPP to characterize the relations among items. By contrast, we aim to generate a relevant and diverse recommendation list through the MAP inference.

The diversity considered in our paper is different from the aggregate diversity in [1, 38]. Increasing aggregate diversity promotes long tail items, while improving diversity prefers diverse items in each recommendation list.

## 3 Fast Greedy MAP Inference

In this section, we present a fast implementation of the greedy MAP inference algorithm for DPP. In each iteration, item

$$j = \arg \max_{i \in Z \setminus Y_\mathrm{g}} \log \det(\mathbf{L}_{Y_\mathrm{g} \cup \{i\}}) - \log \det(\mathbf{L}_{Y_\mathrm{g}}) \tag{1}$$

is added to the already selected item set $Y_\mathrm{g}$. Since $\mathbf{L}$ is a PSD matrix, all of its principal minors are also PSD. Suppose $\det(\mathbf{L}_{Y_\mathrm{g}}) > 0$, and the Cholesky decomposition of $\mathbf{L}_{Y_\mathrm{g}}$ is available as

$$\mathbf{L}_{Y_\mathrm{g}} = \mathbf{V}\mathbf{V}^\top,$$

---
**Algorithm 1** Fast Greedy MAP Inference
---
1: **Input:** Kernel $\mathbf{L}$, *stopping criteria*
2: **Initialize:** $\mathbf{c}_i = []$, $d_i^2 = \mathbf{L}_{ii}$, $j = \arg\max_{i \in Z} \log(d_i^2)$, $Y_{\mathrm{g}} = \{j\}$
3: **while** *stopping criteria* not satisfied **do**
4:     **for** $i \in Z \setminus Y_{\mathrm{g}}$ **do**
5:         $e_i = (\mathbf{L}_{ji} - \langle \mathbf{c}_j, \mathbf{c}_i \rangle)/d_j$
6:         $\mathbf{c}_i = [\mathbf{c}_i \quad e_i]$, $d_i^2 = d_i^2 - e_i^2$
7:     **end for**
8:     $j = \arg\max_{i \in Z \setminus Y_{\mathrm{g}}} \log(d_i^2)$, $Y_{\mathrm{g}} = Y_{\mathrm{g}} \cup \{j\}$
9: **end while**
10: **Return:** $Y_{\mathrm{g}}$
---

where $\mathbf{V}$ is an invertible lower triangular matrix. For any $i \in Z \setminus Y_{\mathrm{g}}$, the Cholesky decomposition of $\mathbf{L}_{Y_{\mathrm{g}} \cup \{i\}}$ can be derived as

$$\mathbf{L}_{Y_{\mathrm{g}} \cup \{i\}} = \begin{bmatrix} \mathbf{L}_{Y_{\mathrm{g}}} & \mathbf{L}_{Y_{\mathrm{g}},i} \\ \mathbf{L}_{i,Y_{\mathrm{g}}} & \mathbf{L}_{ii} \end{bmatrix} = \begin{bmatrix} \mathbf{V} & \mathbf{0} \\ \mathbf{c}_i & d_i \end{bmatrix} \begin{bmatrix} \mathbf{V} & \mathbf{0} \\ \mathbf{c}_i & d_i \end{bmatrix}^{\top}, \tag{2}$$

where row vector $\mathbf{c}_i$ and scalar $d_i \geq 0$ satisfies

$$\mathbf{V}\mathbf{c}_i^{\top} = \mathbf{L}_{Y_{\mathrm{g}},i}, \tag{3}$$

$$d_i^2 = \mathbf{L}_{ii} - \|\mathbf{c}_i\|_2^2. \tag{4}$$

In addition, according to Equ. (2), it can be derived that

$$\det(\mathbf{L}_{Y_{\mathrm{g}} \cup \{i\}}) = \det(\mathbf{V}\mathbf{V}^{\top}) \cdot d_i^2 = \det(\mathbf{L}_{Y_{\mathrm{g}}}) \cdot d_i^2. \tag{5}$$

Therefore, Opt. (1) is equivalent to

$$j = \arg\max_{i \in Z \setminus Y_{\mathrm{g}}} \log(d_i^2). \tag{6}$$

Once Opt. (6) is solved, according to Equ. (2), the Cholesky decomposition of $\mathbf{L}_{Y_{\mathrm{g}} \cup \{j\}}$ becomes

$$\mathbf{L}_{Y_{\mathrm{g}} \cup \{j\}} = \begin{bmatrix} \mathbf{V} & \mathbf{0} \\ \mathbf{c}_j & d_j \end{bmatrix} \begin{bmatrix} \mathbf{V} & \mathbf{0} \\ \mathbf{c}_j & d_j \end{bmatrix}^{\top}, \tag{7}$$

where $\mathbf{c}_j$ and $d_j$ are readily available. The Cholesky factor of $\mathbf{L}_{Y_{\mathrm{g}}}$ can therefore be efficiently updated after a new item is added to $Y_{\mathrm{g}}$.

For each item $i$, $\mathbf{c}_i$ and $d_i$ can also be updated incrementally. After Opt. (6) is solved, define $\mathbf{c}_i'$ and $d_i'$ as the new vector and scalar of $i \in Z \setminus (Y_{\mathrm{g}} \cup \{j\})$. According to Equ. (3) and Equ. (7), we have

$$\begin{bmatrix} \mathbf{V} & \mathbf{0} \\ \mathbf{c}_j & d_j \end{bmatrix} \mathbf{c}_i'^{\top} = \mathbf{L}_{Y_{\mathrm{g}} \cup \{j\},i} = \begin{bmatrix} \mathbf{L}_{Y_{\mathrm{g}},i} \\ \mathbf{L}_{ji} \end{bmatrix}. \tag{8}$$

Combining Equ. (8) with Equ. (3), we conclude

$$\mathbf{c}_i' = [\mathbf{c}_i \quad (\mathbf{L}_{ji} - \langle \mathbf{c}_j, \mathbf{c}_i \rangle)/d_j] \doteq [\mathbf{c}_i \quad e_i].$$

Then Equ. (4) implies

$$d_i'^2 = \mathbf{L}_{ii} - \|\mathbf{c}_i'\|_2^2 = \mathbf{L}_{ii} - \|\mathbf{c}_i\|_2^2 - e_i^2 = d_i^2 - e_i^2. \tag{9}$$

Initially, $Y_{\mathrm{g}} = \emptyset$, and Equ. (5) implies $d_i^2 = \det(\mathbf{L}_{ii}) = \mathbf{L}_{ii}$. The complete algorithm is summarized in Algorithm 1. The *stopping criteria* is $d_j^2 < 1$ for unconstraint MAP inference or $\#Y_{\mathrm{g}} > N$ when the cardinality constraint is imposed. For the latter case, we introduce a small number $\varepsilon > 0$ and add $d_j^2 < \varepsilon$ to the *stopping criteria* for numerical stability of calculating $1/d_j$.

In the $k$-th iteration, for each item $i \in Z \setminus Y_{\mathrm{g}}$, updating $\mathbf{c}_i$ and $d_i$ involve the inner product of two vectors of length $k$, resulting in overall complexity $O(kM)$. Therefore, Algorithm 1 runs in $O(M^3)$ time for unconstraint MAP inference and $O(N^2M)$ to return $N$ items. Notice that this is achieved by additional $O(NM)$ (or $O(M^2)$ for the unconstraint case) space for $\mathbf{c}_i$ and $d_i$.

---

**Algorithm 2** Fast Greedy MAP Inference with a Sliding Window

---
1: **Input:** Kernel $\mathbf{L}$, window size $w$, *stopping criteria*
2: **Initialize:** $\mathbf{V} = []$, $\mathbf{c}_i = []$, $d_i^2 = \mathbf{L}_{ii}$, $j = \arg\max_{i \in Z} \log(d_i^2)$, $Y_g = \{j\}$
3: **while** *stopping criteria* not satisfied **do**
4:      Update $\mathbf{V}$ according to Equ. (7)
5:      **for** $i \in Z \setminus Y_g$ **do**
6:          $e_i = (\mathbf{L}_{ji} - \langle \mathbf{c}_j, \mathbf{c}_i \rangle)/d_j$
7:          $\mathbf{c}_i = [\mathbf{c}_i \quad e_i]$, $d_i^2 = d_i^2 - e_i^2$
8:      **end for**
9:      **if** $\#Y_g \geq w$ **then**
10:          $\mathbf{v} = \mathbf{V}_{2:,1}$, $\mathbf{V} = \mathbf{V}_{2:}$, $a_i = \mathbf{c}_{i,1}$, $\mathbf{c}_i = \mathbf{c}_{i,2:}$
11:          **for** $l = 1, \cdots, w-1$ **do**
12:              $t^2 = \mathbf{V}_{ll}^2 + \mathbf{v}_l^2$
13:              $\mathbf{V}_{l+1:,l} = (\mathbf{V}_{l+1:,l}\mathbf{V}_{ll} + \mathbf{v}_{l+1:}\mathbf{v}_l)/t$, $\mathbf{v}_{l+1:} = (\mathbf{v}_{l+1:}t - \mathbf{V}_{l+1:,l}\mathbf{v}_l)/\mathbf{V}_{ll}$
14:              **for** $i \in Z \setminus Y_g$ **do**
15:                  $\mathbf{c}_{i,l} = (\mathbf{c}_{i,l}\mathbf{V}_{ll} + a_i\mathbf{v}_l)/t$, $a_i = (a_i t - \mathbf{c}_{i,l}\mathbf{v}_l)/\mathbf{V}_{ll}$
16:              **end for**
17:              $\mathbf{V}_{ll} = t$
18:          **end for**
19:          **for** $i \in Z \setminus Y_g$ **do**
20:              $d_i^2 = d_i^2 + a_i^2$
21:          **end for**
22:      **end if**
23:      $j = \arg\max_{i \in Z \setminus Y_g} \log(d_i^2)$, $Y_g = Y_g \cup \{j\}$
24: **end while**
25: **Return:** $Y_g$

---

## 4 Diversity within Sliding Window

In some applications, the selected set of items are displayed as a sequence, and the diversity is only required within a sliding window. Denote the window size as $w$. We modify Opt. (1) to

$$j = \arg\max_{i \in Z \setminus Y_g} \log \det(\mathbf{L}_{Y_g^w \cup \{i\}}) - \log \det(\mathbf{L}_{Y_g^w}), \tag{10}$$

where $Y_g^w \subseteq Y_g$ contains $w-1$ most recently added items. When $\#Y_g \geq w$, a simple modification of method [32] solves Opt. (10) with complexity $O(w^2 M)$. We adapt our algorithm to this scenario so that Opt. (10) can be solved in $O(wM)$ time.

In Section 3, we showed how to efficiently select a new item when $\mathbf{V}$, $\mathbf{c}_i$, and $d_i$ are available. For Opt. (10), $\mathbf{V}$ is the Cholesky factor of $\mathbf{L}_{Y_g^w}$. After Opt. (10) is solved, we can similarly update $\mathbf{V}$, $\mathbf{c}_i$, and $d_i$ for $\mathbf{L}_{Y_g^w \cup \{j\}}$. When the number of items in $Y_g^w$ is $w-1$, to update $Y_g^w$, we also need to remove the earliest added item in $Y_g^w$. The detailed derivations of updating $\mathbf{V}$, $\mathbf{c}_i$, and $d_i$ when the earliest added item is removed are given in the supplementary material.

The complete algorithm is summarized in Algorithm 2. Line 10-21 shows how to update $\mathbf{V}$, $\mathbf{c}_i$, and $d_i$ in place after the earliest item is removed. In the $k$-th iteration where $k \geq w$, updating $\mathbf{V}$, all $\mathbf{c}_i$ and $d_i$ require $O(w^2)$, $O(wM)$, and $O(M)$ time, respectively. The overall complexity of Algorithm 2 is $O(wNM)$ to return $N \geq w$ items. Numerical stability is discussed in the supplementary material.

## 5 Improving Recommendation Diversity

In this section, we describe a DPP-based approach for recommending relevant and diverse items to users. For a user $u$, the profile item set $P_u$ is defined as the set of items that the user likes. Based on $P_u$, a recommender system recommends items $R_u$ to the user.

The approach takes three inputs: a candidate item set $C_u$, a score vector $\mathbf{r}_u$ which indicates how relevant the items in $C_u$ are, and a PSD matrix $\mathbf{S}$ which measures the similarity of each pair of items. The first two inputs can be obtained from the internal results of many traditional recommendation

algorithms. The third input, similarity matrix $\mathbf{S}$, can be obtained based on the attributes of items, the interaction relations with users, or a combination of both. This approach can be regarded as a ranking algorithm balancing the relevance of items and their similarities.

To apply the DPP model in the recommendation task, we need to construct the kernel matrix. As revealed in [30], the kernel matrix can be written as a Gram matrix, $\mathbf{L} = \mathbf{B}^\top \mathbf{B}$, where the columns of $\mathbf{B}$ are vectors representing the items. We can construct each column vector $\mathbf{B}_i$ as the product of the item score $r_i \geq 0$ and a normalized vector $\mathbf{f}_i \in \mathbb{R}^D$ with $\|\mathbf{f}_i\|_2 = 1$. The entries of kernel $\mathbf{L}$ can be written as

$$\mathbf{L}_{ij} = \langle \mathbf{B}_i, \mathbf{B}_j \rangle = \langle r_i \mathbf{f}_i, r_j \mathbf{f}_j \rangle = r_i r_j \langle \mathbf{f}_i, \mathbf{f}_j \rangle. \tag{11}$$

We can think of $\langle \mathbf{f}_i, \mathbf{f}_j \rangle$ as measuring the similarity between item $i$ and item $j$, i.e., $\langle \mathbf{f}_i, \mathbf{f}_j \rangle = \mathbf{S}_{ij}$. Therefore, the kernel matrix for user $u$ can be written as

$$\mathbf{L} = \mathrm{Diag}(\mathbf{r}_u) \cdot \mathbf{S} \cdot \mathrm{Diag}(\mathbf{r}_u),$$

where $\mathrm{Diag}(\mathbf{r}_u)$ is a diagonal matrix whose diagonal vector is $\mathbf{r}_u$. The log-probability of $R_u$ is

$$\log \det(\mathbf{L}_{R_u}) = \sum_{i \in R_u} \log(\mathbf{r}_{u,i}^2) + \log \det(\mathbf{S}_{R_u}). \tag{12}$$

The second term in Equ. (12) is maximized when the item representations of $R_u$ are orthogonal, and therefore it promotes diversity. It clearly shows how the DPP model incorporates the relevance and diversity of the recommended items.

A nice feature of methods in [11, 51, 8] is that they involve a tunable parameter which allows users to adjust the trade-off between relevance and diversity. According to Equ. (12), the original DPP model does not offer such a mechanism. We modify the log-probability of $R_u$ to

$$\log \mathcal{P}(R_u) \propto \theta \cdot \sum_{i \in R_u} \mathbf{r}_{u,i} + (1 - \theta) \cdot \log \det(\mathbf{S}_{R_u}),$$

where $\theta \in [0, 1]$. This corresponds to a DPP with kernel

$$\mathbf{L}' = \mathrm{Diag}(\exp(\alpha \mathbf{r}_u)) \cdot \mathbf{S} \cdot \mathrm{Diag}(\exp(\alpha \mathbf{r}_u)),$$

where $\alpha = \theta/(2(1 - \theta))$. We can also get the marginal gain of log-probability

$$\log \mathcal{P}(R_u \cup \{i\}) - \log \mathcal{P}(R_u) \propto \theta \cdot \mathbf{r}_{u,i} + (1 - \theta) \cdot (\log \det(\mathbf{S}_{R_u \cup \{i\}}) - \log \det(\mathbf{S}_{R_u})). \tag{13}$$

Then Algorithm 1 and Algorithm 2 can be easily modified to maximize (13) with kernel matrix $\mathbf{S}$.

Notice that we need the similarity $\mathbf{S}_{ij} \in [0, 1]$ for the recommendation task, where 0 means the most diverse and 1 means the most similar. This may be violated when the inner product of normalized vectors $\langle \mathbf{f}_i, \mathbf{f}_j \rangle$ can take negative values. In the extreme case, the most diverse pair $\mathbf{f}_i = -\mathbf{f}_j$, but the determinant of the corresponding sub-matrix is 0, same as $\mathbf{f}_i = \mathbf{f}_j$. To guarantee nonnegativity, we can take a linear mapping while keeping $\mathbf{S}$ a PSD matrix, e.g.,

$$\mathbf{S}_{ij} = \frac{1 + \langle \mathbf{f}_i, \mathbf{f}_j \rangle}{2} = \left\langle \frac{1}{\sqrt{2}} \begin{bmatrix} 1 \\ \mathbf{f}_i \end{bmatrix}, \frac{1}{\sqrt{2}} \begin{bmatrix} 1 \\ \mathbf{f}_j \end{bmatrix} \right\rangle \in [0, 1].$$

## 6 Experimental Results

In this section, we evaluate and compare our proposed algorithms on synthetic dataset and real-world recommendation tasks. Algorithms are implemented in Python with vectorization. The experiments are performed on a laptop with 2.2GHz Intel Core i7 and 16GB RAM.

### 6.1 Synthetic Dataset

In this subsection, we evaluate the performance of our Algorithm 1 on the MAP inference for DPP. We follow the experimental setup in [20]. The entries of the kernel matrix satisfy Equ. (11), where $r_i = \exp(0.01 x_i + 0.2)$ with $x_i \in \mathbb{R}$ drawn from the normal distribution $\mathcal{N}(0, 1)$, and $\mathbf{f}_i \in \mathbb{R}^D$ with $D$ same as the total item size $M$ and entries drawn i.i.d. from $\mathcal{N}(0, 1)$ and then normalized.

Our proposed faster exact algorithm (FaX) is compared with Schur complement combined with lazy evaluation (Lazy) [36] and faster approximate algorithm (ApX) [20]. The parameters of the reference

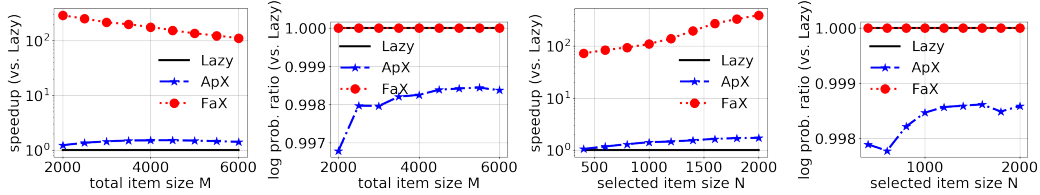

Figure 1: Comparison of Lazy, ApX, and our FaX under different $M$ when $N = 1000$ (left), and under different $N$ when $M = 6000$ (right).

algorithms are chosen as suggested in [20]. The gradient-based method in [17] and the double greedy algorithm in [10] are not compared because as reported in [20], they performed worse than ApX. We report the speedup over Lazy of each algorithm, as well as the ratio of log-probability [20]

$$\log \det \mathbf{L}_Y / \log \det \mathbf{L}_{Y_{\text{Lazy}}},$$

where $Y$ and $Y_{\text{Lazy}}$ are the outputs of an algorithm and Lazy, respectively. We compare these metrics when the total item size $M$ varies from 2000 to 6000 with the selected item size $N = 1000$, and when $N$ varies from 400 to 2000 with $M = 6000$. The results are averaged over 10 independent trials, and shown in Figure 1. In both cases, FaX runs significantly faster than ApX, which is the state-of-the-art fast greedy MAP inference algorithm for DPP. FaX is about 100 times faster than Lazy, while ApX is about 3 times faster, as reported in [20]. The accuracy of FaX is the same as Lazy, because they are exact implementations of the greedy algorithm. ApX loses about $0.2\%$ accuracy.

## 6.2 Short Sequence Recommendation

In this subsection, we evaluate the performance of Algorithm 1 to recommend short sequences of items to users on the following two public datasets.

**Netflix Prize**[2]: This dataset contains users' ratings of movies. We keep ratings of four or higher and binarize them. We only keep users who have watched at least 10 movies and movies that are watched by at least 100 users. This results in $436,674$ users and $11,551$ movies with $56,406,404$ ratings.

**Million Song Dataset** [6]: This dataset contains users' play counts of songs. We binarize play counts of more than once. We only keep users who listen to at least 20 songs and songs that are listened to by at least 100 users. This results in $213,949$ users and $20,716$ songs with $8,500,651$ play counts.

For each dataset, we construct the test set by randomly selecting one interacted item for each user, and use the rest data for training. We adopt an item-based recommendation algorithm [24] on the training set to learn an item-item PSD similarity matrix $\mathbf{S}$. For each user, the profile set $P_u$ consists of the interacted items in the training set, and the candidate set $C_u$ is formed by the union of 50 most similar items of each item in $P_u$. The median of $\#C_u$ is 735 and 811 on Netflix Prize and Million Song Dataset, respectively. For any item in $C_u$, the relevance score is the aggregated similarity to all items in $P_u$ [22]. With $\mathbf{S}$, $C_u$, and the score vector $\mathbf{r}_u$, algorithms recommend $N = 20$ items.

Performance metrics of recommendation include mean reciprocal rank (MRR) [46], intra-list average distance (ILAD) [51], and intra-list minimal distance (ILMD). They are defined as

$$\text{MRR} = \underset{u \in U}{\text{mean}}\, p_u^{-1}, \quad \text{ILAD} = \underset{u \in U}{\text{mean}}\, \underset{i,j \in R_u, i \neq j}{\text{mean}} (1 - \mathbf{S}_{ij}), \quad \text{ILMD} = \underset{u \in U}{\text{mean}}\, \underset{i,j \in R_u, i \neq j}{\min} (1 - \mathbf{S}_{ij}),$$

where $U$ is the set of all users, and $p_u$ is the smallest rank position of items in the test set. MRR measures relevance, while ILAD and ILMD measure diversity. We also compare the metric popularity-weighted recall (PW Recall) [42] in the supplementary material. For these metrics, higher ones are preferred.

Our DPP-based algorithm (DPP) is compared with maximal marginal relevance (MMR) [11], max-sum diversification (MSD) [8], entropy regularizer (Entropy) [40], and coverage-based algorithm (Cover) [39]. They all involve a tunable parameter to adjust the trade-off between relevance and diversity. For Cover, the parameter is $\gamma \in [0, 1]$ which defines the saturation function $f(t) = t^\gamma$.

In the first experiment, we test the impact of trade-off parameter $\theta \in [0, 1]$ of DPP on Netflix Prize dataset. The results are shown in Figure 2. As $\theta$ increases, MRR improves at first, achieves the best

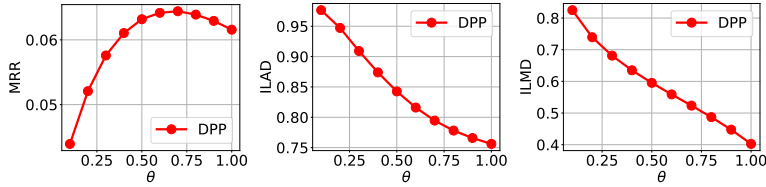

Figure 2: Impact of trade-off parameter $\theta$ on Netflix Prize dataset.

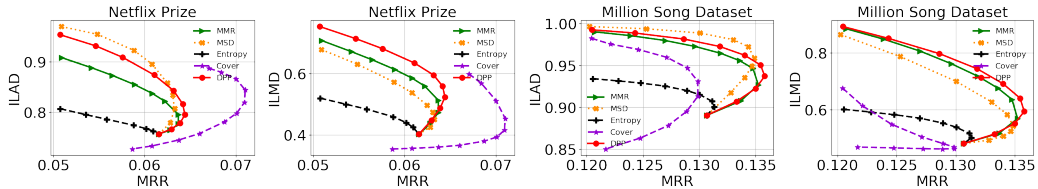

Figure 3: Comparison of trade-off performance between relevance and diversity under different choices of trade-off parameters on Netflix Prize (left) and Million Song Dataset (right). The standard error of MRR is about $0.0003$ and $0.0006$ on Netflix Prize and Million Song Dataset, respectively.

Table 1: Comparison of average / upper $99\%$ running time (in milliseconds).

| Dataset | MMR | MSD | Entropy | Cover | DPP |
| --- | --- | --- | --- | --- | --- |
| Netflix Prize | 0.23 / 0.50 | 0.21 / 0.41 | 200.74 / 2883.82 | 120.19 / 1332.21 | 0.73 / 1.75 |
| Million Song Dataset | 0.23 / 0.41 | 0.22 / 0.34 | 26.45 / 168.12 | 23.76 / 173.64 | 0.76 / 1.46 |

value when $\theta \approx 0.7$, and then decreases a little bit. ILAD and ILMD are monotonously decreasing as $\theta$ increases. When $\theta = 1$, DPP returns items with the highest relevance scores. Therefore, taking moderate amount of diversity into consideration, better performance can be achieved.

In the second experiment, by varying the trade-off parameters, the trade-off performance between relevance and diversity are compared in Figure 3. The parameters are chosen such that different algorithms have approximately the same range of MRR. As can be seen, Cover performs the best on Netflix Prize but becomes the worst on Million Song Dataset. Among the other algorithms, DPP enjoys the best relevance-diversity trade-off performance. Their average and upper $99\%$ running time are compared in Table 1. MMR, MSD, and DPP run significantly faster than Entropy and Cover. Since DPP runs in less than 2ms with probability $99\%$, it can be used in real-time scenarios.

We conducted an online A/B test in a movie recommender system for four weeks. For each user, candidate movies with relevance scores were generated by an online scoring model. An offline matrix factorization algorithm [26] was trained daily to generate movie representations which were used to calculate similarities. For the control group, $5\%$ users were randomly selected and presented with $N = 8$ movies with the highest relevance scores. For the treatment group, another $5\%$ random users were presented with $N$ movies generated by DPP with a fine-tuned trade-off parameter. Two online metrics, improvements of number of titles watched and watch minutes, are reported in Table 2. The results are also compared with another $5\%$ randomly selected users using MMR. As can be seen, DPP performed better compared with systems without diversification algorithm or with MMR.

### 6.3 Long Sequence Recommendation

In this subsection, we evaluate the performance of Algorithm 2 to recommend long sequences of items to users. For each dataset, we construct the test set by randomly selecting 5 interacted items for each user, and use the rest for training. Each long sequence contains $N = 100$ items. We choose window size $w = 10$ so that every $w$ successive items in the sequence are diverse. The value of $w$ usually depends on specific applications. Generally speaking, if each time a user can only see a small portion of the sequence, $w$ can be of the order of the portion size. Other settings are the same as in the previous subsection.

Table 2: Performance improvement of MMR and DPP over Control in an online A/B test.

| Algorithm | Improvement of No. Titles Watched | Improvement of Watch Minutes |
|---|---|---|
| MMR | 0.84% | 0.86% |
| DPP | 1.33% | 1.52% |

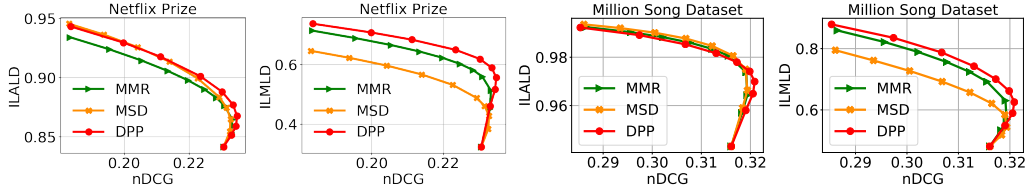

Figure 4: Comparison of trade-off performance between relevance and diversity under different choices of trade-off parameters on Netflix Prize (left) and Million Song Dataset (right). The standard error of nDCG is about $0.00025$ and $0.0005$ on Netflix Prize and Million Song Dataset, respectively.

Performance metrics include normalized discounted cumulative gain (nDCG) [23], intra-list average local distance (ILALD), and intra-list minimal local distance (ILMLD). The latter two are defined as

$$\text{ILALD} = \underset{u \in U}{\text{mean}} \ \underset{i,j \in R_u, i \neq j, d_{ij} \leq w}{\text{mean}} (1 - \mathbf{S}_{ij}), \quad \text{ILMLD} = \underset{u \in U}{\text{mean}} \ \underset{i,j \in R_u, i \neq j, d_{ij} \leq w}{\min} (1 - \mathbf{S}_{ij}),$$

where $d_{ij}$ is the position distance of item $i$ and $j$ in $R_u$. Similarly, higher metrics are desirable. To make a fair comparison, we modify the diversity terms in MMR and MSD so that they only consider the most recently added $w - 1$ items. Entropy and Cover are not tested because they are not suitable for this scenario. By varying trade-off parameters, the trade-off performance between relevance and diversity of MMR, MSD, and DPP are compared in Figure 4. The parameters are chosen such that different algorithms have approximately the same range of nDCG. As can be seen, DPP performs the best with respect to relevance-diversity trade-off. We also compare the metric PW Recall in the supplementary material.

## 7  Conclusion and Future Work

In this paper, we presented a fast and exact implementation of the greedy MAP inference for DPP. The time complexity of our algorithm is $O(M^3)$, which is significantly lower than state-of-the-art exact implementations. Our proposed acceleration technique can be applied to other problems with log-determinant of PSD matrices in the objective functions, such as the entropy regularizer [40]. We also adapted our fast algorithm to scenarios where the diversity is only required within a sliding window. Experiments showed that our algorithm runs significantly faster than state-of-the-art algorithms, and our proposed approach provides better relevance-diversity trade-off on recommendation task. Potential future research directions include learning the optimal trade-off parameter automatically and the theoretical analysis of Algorithm 2.

### Acknowledgments

We thank Bangsheng Tang, Yin Zheng, Shenglong Lv, and the reviewers for many helpful discussions and suggestions.

## Footnotes

[2]Netflix Prize website, http://www.netflixprize.com/

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
