[Supplementary Material]

# Supplementary Material

## A    Update of $\mathbf{V}$, $\mathbf{c}_i$, and $d_i$

Hereinafter, we use $Y_{\mathrm{g}}^w$ to denote the set after the earliest added item $h$ is removed. We will show how to update $\mathbf{V}$, $\mathbf{c}_i$, and $d_i$ when $h$ is removed from $\{h\} \cup Y_{\mathrm{g}}^w$.

### A.1    Update of $\mathbf{V}$

Since the Cholesky factor of $\mathbf{L}_{\{h\} \cup Y_{\mathrm{g}}^w}$ is $\mathbf{V}$,

$$\begin{bmatrix} \mathbf{L}_{hh} & \mathbf{L}_{h,Y_{\mathrm{g}}^w} \\ \mathbf{L}_{Y_{\mathrm{g}}^w,h} & \mathbf{L}_{Y_{\mathrm{g}}^w} \end{bmatrix} = \begin{bmatrix} \mathbf{V}_{11} & \mathbf{0} \\ \mathbf{V}_{2:,1} & \mathbf{V}_{2:} \end{bmatrix} \begin{bmatrix} \mathbf{V}_{11} & \mathbf{0} \\ \mathbf{V}_{2:,1} & \mathbf{V}_{2:} \end{bmatrix}^\top, \tag{14}$$

where $2\!:$ denotes the indices starting from 2 to the end. For simplicity, hereinafter we use $\mathbf{v} = \mathbf{V}_{2:,1}$ and $\ddot{\mathbf{V}} = \mathbf{V}_{2:}$. Let $\mathbf{V}'$ denote the Cholesky factor of $\mathbf{L}_{Y_{\mathrm{g}}^w}$. Equ. (14) implies

$$\mathbf{V}'\mathbf{V}'^\top = \mathbf{L}_{Y_{\mathrm{g}}^w} = \ddot{\mathbf{V}}\ddot{\mathbf{V}}^\top + \mathbf{v}\mathbf{v}^\top. \tag{15}$$

Since $\mathbf{V}'$ and $\ddot{\mathbf{V}}$ are lower triangular matrices of the same size, Equ. (15) is a classical rank-one update for Cholesky decomposition. We follow the procedure described in [41] for this problem. Let

$$\mathbf{V}' = \begin{bmatrix} \mathbf{V}'_{11} & \mathbf{0} \\ \mathbf{V}'_{2:,1} & \mathbf{V}'_{2:} \end{bmatrix}, \; \ddot{\mathbf{V}} = \begin{bmatrix} \ddot{\mathbf{V}}_{11} & \mathbf{0} \\ \ddot{\mathbf{V}}_{2:,1} & \ddot{\mathbf{V}}_{2:} \end{bmatrix}, \; \mathbf{v} = \begin{bmatrix} \mathbf{v}_1 \\ \mathbf{v}_{2:} \end{bmatrix}.$$

Then Equ. (15) is equivalent to

$$\mathbf{V}'^2_{11} = \ddot{\mathbf{V}}^2_{11} + \mathbf{v}^2_1, \tag{16}$$

$$\mathbf{V}'_{2:,1}\mathbf{V}'_{11} = \ddot{\mathbf{V}}_{2:,1}\ddot{\mathbf{V}}_{11} + \mathbf{v}_{2:}\mathbf{v}_1,$$

$$\mathbf{V}'_{2:}\mathbf{V}'^\top_{2:} + \mathbf{V}'_{2:,1}\mathbf{V}'^\top_{2:,1} = \ddot{\mathbf{V}}_{2:}\ddot{\mathbf{V}}^\top_{2:} + \ddot{\mathbf{V}}_{2:,1}\ddot{\mathbf{V}}^\top_{2:,1} + \mathbf{v}_{2:}\mathbf{v}^\top_{2:}$$

which imply

$$\mathbf{V}'_{2:,1} = (\ddot{\mathbf{V}}_{2:,1}\ddot{\mathbf{V}}_{11} + \mathbf{v}_{2:}\mathbf{v}_1)/\mathbf{V}'_{11}, \tag{17}$$

$$\mathbf{V}'_{2:}\mathbf{V}'^\top_{2:} = \ddot{\mathbf{V}}_{2:}\ddot{\mathbf{V}}^\top_{2:} + \mathbf{v}'\mathbf{v}'^\top, \tag{18}$$

$$\mathbf{v}' = (\mathbf{v}_{2:}\mathbf{V}'_{11} - \mathbf{V}'_{2:,1}\mathbf{v}_1)/\ddot{\mathbf{V}}_{11}. \tag{19}$$

The first column of $\mathbf{V}'$ can be determined by Equ. (16) and Equ. (17). For the rest part, notice that Equ. (18) together with Equ. (19) is again a rank-one update but with a smaller size. We can repeat the aforementioned procedure until the last diagonal element of $\mathbf{V}'$ is obtained.

### A.2    Update of $\mathbf{c}_i$

According to Equ. (3),

$$\begin{bmatrix} \mathbf{V}_{11} & \mathbf{0} \\ \mathbf{v} & \ddot{\mathbf{V}} \end{bmatrix} \begin{bmatrix} \mathbf{c}_{i,1} & \mathbf{c}_{i,2:} \end{bmatrix}^\top = \begin{bmatrix} \mathbf{L}_{hi} \\ \mathbf{L}_{Y_{\mathrm{g}}^w,i} \end{bmatrix},$$

where $\mathbf{c}_{i,1}$ denotes the first element of $\mathbf{c}_i$ and $\mathbf{c}_{i,2:}$ is the remaining sub-vector. Let $a_i = \mathbf{c}_{i,1}$ and $\ddot{\mathbf{c}}_i = \mathbf{c}_{i,2:}$. Define $\mathbf{c}'_i$ as the vector of item $i$ after $h$ is removed. Then

$$\mathbf{V}'\mathbf{c}'^\top_i = \mathbf{L}_{Y_{\mathrm{g}}^w,i} = \ddot{\mathbf{V}}\ddot{\mathbf{c}}^\top_i + \mathbf{v}a_i. \tag{20}$$

Let

$$\mathbf{c}'_i = \begin{bmatrix} \mathbf{c}'_{i,1} & \mathbf{c}'_{i,2:} \end{bmatrix}, \; \ddot{\mathbf{c}}_i = \begin{bmatrix} \ddot{\mathbf{c}}_{i,1} & \ddot{\mathbf{c}}_{i,2:} \end{bmatrix}.$$

Then Equ. (20) is equivalent to

$$\mathbf{c}'_{i,1}\mathbf{V}'_{11} = \ddot{\mathbf{c}}_{i,1}\ddot{\mathbf{V}}_{11} + a_i\mathbf{v}_1,$$

$$\mathbf{V}'_{2:}\mathbf{c}'^\top_{i,2:} + \mathbf{V}'_{2:,1}\mathbf{c}'_{i,1} = \ddot{\mathbf{V}}_{2:}\ddot{\mathbf{c}}^\top_{i,2:} + \ddot{\mathbf{V}}_{2:,1}\ddot{\mathbf{c}}_{i,1} + \mathbf{v}_{2:}a_i$$

which imply

$$\mathbf{c}'_{i,1} = (\ddot{\mathbf{c}}_{i,1}\ddot{\mathbf{V}}_{11} + a_i\mathbf{v}_1)/\mathbf{V}'_{11}, \tag{21}$$

$$\mathbf{V}'_{2:}\mathbf{c}'^{\top}_{i,2:} = \ddot{\mathbf{V}}_{2:}\ddot{\mathbf{c}}^{\top}_{i,2:} + \mathbf{v}'a'_i, \tag{22}$$

$$a'_i = (a_i\mathbf{V}'_{11} - \mathbf{c}'_{i,1}\mathbf{v}_1)/\ddot{\mathbf{V}}_{11}. \tag{23}$$

The first element of $\mathbf{c}'_i$ can be determined by Equ. (21). For the rest part, since Equ. (22) together with Equ. (19) and (23) has the same form as Equ. (20), we can repeat the aforementioned procedure until we get the last element of $\mathbf{c}'_i$.

### A.3  Update of $d_i$

According to Equ. (4),

$$d_i^2 = \mathbf{L}_{ii} - a_i^2 - \|\ddot{\mathbf{c}}_i\|_2^2.$$

Define $d'_i$ as the scalar of item $i$ after $h$ is removed. Then

$$
\begin{aligned}
d_i'^2 &= \mathbf{L}_{ii} - \|\mathbf{c}'_i\|_2^2 \\
&= d_i^2 + a_i^2 + \|\ddot{\mathbf{c}}_i\|_2^2 - \|\mathbf{c}'_i\|_2^2 \\
&= d_i^2 + a_i^2 + \ddot{\mathbf{c}}_{i,1}^2 + \|\ddot{\mathbf{c}}_{i,2:}\|_2^2 - \mathbf{c}'^2_{i,1} - \|\mathbf{c}'_{i,2:}\|_2^2 \\
&= d_i^2 + a_i'^2 + \|\ddot{\mathbf{c}}_{i,2:}\|_2^2 - \|\mathbf{c}'_{i,2:}\|_2^2
\end{aligned}
$$

$$\tag{24}$$
$$\tag{25}$$

where Equ. (25) is due to

$$
\begin{aligned}
a_i^2 + \ddot{\mathbf{c}}_{i,1}^2 - \mathbf{c}'^2_{i,1} &\stackrel{\text{Equ.}(21)}{=\!=\!=\!=\!=} a_i^2 + (\mathbf{c}'_{i,1}\mathbf{V}'_{11} - a_i\mathbf{v}_1)^2/\ddot{\mathbf{V}}_{11}^2 - \mathbf{c}'^2_{i,1} \\
&= (a_i^2(\ddot{\mathbf{V}}_{11}^2 + \mathbf{v}_1^2) + \mathbf{c}'^2_{i,1}(\mathbf{V}'^2_{11} - \ddot{\mathbf{V}}_{11}^2) - 2\mathbf{c}'_{i,1}\mathbf{V}'_{11}a_i\mathbf{v}_1)/\ddot{\mathbf{V}}_{11}^2 \\
&\stackrel{\text{Equ.}(16)}{=\!=\!=\!=\!=} (a_i^2\mathbf{V}'^2_{11} + \mathbf{c}'^2_{i,1}\mathbf{v}_1^2 - 2\mathbf{c}'_{i,1}\mathbf{V}'_{11}a_i\mathbf{v}_1)/\ddot{\mathbf{V}}_{11}^2 \\
&\stackrel{\text{Equ.}(23)}{=\!=\!=\!=\!=} a_i'^2.
\end{aligned}
$$

Notice that Equ. (25) has the same form as Equ. (24). Therefore, after $\mathbf{c}'_i$ has been updated, we can directly get $d_i'^2 = d_i^2 + a_i'^2$.

## B  Discussion on Numerical Stability

As introduced in Section 3, updating $\mathbf{c}_i$ and $d_i$ involves calculating $e_i$, where $d_j^{-1}$ is involved. If $d_j$ is approximately zero, our algorithm encounters the numerical instability issue. According to Equ. (5), $d_j$ satisfies

$$d_j^2 = \frac{\det(\mathbf{L}_{Y_g \cup \{j\}})}{\det(\mathbf{L}_{Y_g})}. \tag{26}$$

Let $j_k$ be the selected item in the $k$-th iteration. Theorem 1 gives some results about the sequence $\{d_{j_k}\}$.

**Theorem 1.** *In Algorithm 1, $\{d_{j_k}\}$ is non-increasing, and $d_{j_k} > 0$ if and only if $k \leq \mathrm{rank}(\mathbf{L})$.*

*Proof.* First, in the $k$-th iteration, since $j_k$ is the solution to Opt. (6), $d_{j_k} \geq d_{j_{k+1}}$. After $j_k$ is added, $d_{j_{k+1}}$ does not increase after update Equ. (9). Therefore, sequence $\{d_{j_k}\}$ is non-increasing.

Now we prove the second part of the theorem. Let $V_k \subseteq Z$ be the items that have been selected by Algorithm 1 at the end of the $k$-th step. Let $W_k \subseteq Z$ be a set of $k$ items such that $\det(\mathbf{L}_{W_k})$ is maximum. According to Theorem 3.3 in [13], we have

$$\det(\mathbf{L}_{V_k}) \geq \left(\frac{1}{k!}\right)^2 \cdot \det(\mathbf{L}_{W_k}).$$

When $k \leq \mathrm{rank}(\mathbf{L})$, $W_k$ satisfies $\det(\mathbf{L}_{W_k}) > 0$, and therefore

$$\det(\mathbf{L}_{V_k}) > 0, \quad k \leq \mathrm{rank}(\mathbf{L}).$$

Figure 5: Impact of trade-off parameter $\theta$ on PW Recall on Netflix dataset.

Figure 6: Comparison of trade-off performance between MRR and PW Recall under different choices of trade-off parameters on Netflix Prize (left) and Million Song Dataset (right).

Figure 7: Comparison of trade-off performance between nDCG and PW Recall under different choices of trade-off parameters on Netflix Prize (left) and Million Song Dataset (right).

According to Equ. (26), we have

$$d_{j_k}^2 = \frac{\det(\mathbf{L}_{V_k})}{\det(\mathbf{L}_{V_{k-1}})}.$$

As a result, for $k = 1, \ldots, \mathrm{rank}(\mathbf{L})$, $d_{j_k} > 0$. When $k = \mathrm{rank}(\mathbf{L}) + 1$, $V_k$ contains $\mathrm{rank}(\mathbf{L}) + 1$ items, and $\mathbf{L}_{V_k}$ is singular. Therefore, $d_{j_k} = 0$ for $k = \mathrm{rank}(\mathbf{L}) + 1$. $\qquad\square$

According to Theorem 1, when kernel $\mathbf{L}$ is a low rank matrix, Algorithm 1 returns at most $\mathrm{rank}(\mathbf{L})$ items. For Algorithm 2 with a sliding window, according to Subsection A.3, $d_i$ is non-decreasing after the earliest item is removed. This allows for returning more items, and alleviates the numerical instability issue.

## C  More Simulation Results

We have also compared the metric popularity-weighted recall (PW Recall) [42] of different algorithms. Its definition is

$$\mathrm{PW\ Recall} = \frac{\sum_{u \in U} \sum_{t \in T_u} w_t \mathbb{I}_{t \in R_u}}{\sum_{u \in U} \sum_{t \in T_u} w_t},$$

where $T_u$ is the set of relevant items in the test set, $w_t$ is the weight of item $t$ with $w_t \propto C(t)^{-0.5}$ where $C(t)$ is the number of occurrences of $t$ in the training set, and $\mathbb{I}_P$ is the indicator function. PW Recall measures both relevance and diversity.

Similar to Figure 2, the impact of trade-off parameter $\theta$ on PW Recall on Netflix Prize is shown in Figure 5. As $\theta$ increases, MRR improves at first, achieves the best value when $\theta \approx 0.5$, and then decreases. Therefore, moderate amount of diversity also leads to better PW Recall.

Similar to Figure 3, by varying the trade-off parameters, the trade-off performance between MRR and PW Recall are compared in Figure 6. Similar conclusions can be drawn.

Similar to Figure 4, by varying the trade-off parameters, the trade-off performance between nDCG and PW Recall are compared in Figure 7. DPP enjoys the best trade-off performance on both datasets.