[Reviews · NeurIPS 2018]

Reviewer 1



This paper proposes fast approximation of the maximum a posteriori (MAP) inference for determinantal point processes (DPPs), which is NP-hard problem. The submodular nature allows greedy algorithm to approximate DPP MAP, but its direct computation has expensive complexity. In this work, authors provide efficient greedy DPP MAP inference algorithm using Cholesky factored properties. In other words, every greedy update requires marginal gain of log-determinant and this can be exactly computed using only vector inner products. Therefore, the computational complexity can be improved to O(M^3) for given M ground sets (the naïve implementation is O(M^5)). Practically, it runs much faster than the state-of-the-art DPP MAP inference approximations. Authors also provide the algorithm for diverse sequence using sliding window approach without hurt of algorithm complexity. In their experiments, the proposed algorithms show the best results on sequence recommendation under real-world dataset. In overall, this paper is very nice work as it adopts a novel idea for fast and efficient computations and shows better practical speedup and accuracy than the state-of-the-art algorithm. I believe that this paper is enough to accept to NIPS. One drawback is a lack of rigorous analysis. For example, authors could mention at least approximation guarantee (i.e., 1-1/e) of the general greedy algorithm. I am wondering the theoretic guarantee of sliding window version of greedy algorithm (Algorithm 2). When w=1, it corresponds to Algorithm 1, but how user could decide the value of w? The minor comments: The description of algorithm 1 is somewhat confusing, e.g., is c_i increases over the both inner and outer iterations? Authors should modify the notations. It seems that the proposed algorithms can adopt lazy evaluations. It is expected that lazy evaluations boosts the algorithms even faster. Authors can clarify the possibility of adaption of lazy evaluations in their algorithms.

Reviewer 2



The goal of this paper is to speed up the greedy MAP inference for determinantal point process (DPP). The proposed algorithm updates the Cholesky facto r incrementally which leads to a lower time complexity compared to the state-of-the-art algorithms. The proposed algorithm is also verified by online A/B testing and is proven to achieve a better relevance and diversity tradeoff. Overall speaking, the main contribution of this paper is the lower time complexity compared to other greedy algorithms for MAP inference, although most of the techniques are standard. The application is interesting. In your long sequence recommendation experiment part, you choose the window length to be w=10. Is your experiment sensitive to the choice of window length? In practice, how to choose w?

Reviewer 3



Summary: This paper introduces an exact algorithm for greedy mode finding for DPPs which is faster by a factor of M (ground set size) than previous work on greedy MAP algorithms for DPPs; the authors also show that this algorithm can be further sped up when diversity is required over only a sliding window within long recommendations. As an additional contribution, the authors show that modeling recommendation problems with DPPs and generating recommendations via their algorithm outperforms other standard (non-DPP) recommender algorithms along various metrics. ------------------- This paper builds upon the recent growing interest in leveraging Determinantal Point Processes for various machine learning problems. As the authors mention, a key advantage of DPPs is their ability to tractably balance quality and diversity requirements for most operations, with mode estimation being one of the only operations that remains NP-hard. Indeed, sampling from a DPP has been used in previous literature, presumably as a more scalable alternative to greedy MAP finding (e.g. for network compression). Although the usefulness of DPPs for recommender systems is now an accepted fact, the analysis provided in section 5 and 6.2 remains interesting, in particular thanks to the discussion of the tunable scaling of diversity and quality preferences and how it can easily be incorporated into the new formulation of the greedy algorithm. However, I would like to see a detailed theoretical comparison of the proposed greedy algorithm to the derandomized version of the lazy DPP sampling algorithm proposed in the thesis (Gillenwater, 2014). -------------------- Quality: This paper is of good quality. Clarity: This paper is very clear. Originality: This paper is marginally novel - see below regarding its potential similarities to prior work. To my knowledge, however, there has not been an explicit mention of a fast greedy algorithm for DPP sampling prior to this submission. Significance: Fast greedy algorithms for DPPs are significant, as well as more broadly within recommendation systems (although the M^3 complexity might be a more general impediment to the use of the greedy MAP algorithm for DPPs within recommendations over very large set sizes). -------------------- Detailed comments: - Could you please detail the difference between your proposed algorithm and a derandomized greedy version of Alg. 2 of "Approximate Inference for DPPs" (Gillenwater, 2014) that uses all eigenvalues for the variable set V? In particular, the initialization of your variables d_i^2 is the same as that of the z_i of Gillenwater's Alg. 2, and the update rules also seem similar (although they are derived using a different logic). - You might want to explicitly mention that higher MRR/ILAD/... are desirable for readers who aren't familiar with recommender systems. - Could you provide some intuition as to how the greedy algorithm performs when compared to simple DPP sampling for recommendations? It would be interesting to see how much of a gain the greedy approach provides over just sampling multiple times (O(M^3) if you need to compute the kernel eigendecomposition) from the DPP and keeping the subset with the highest probability.